# Did the Brain and Oral Microbiota Talk to Each Other? A Review of the Literature

**DOI:** 10.3390/jcm9123876

**Published:** 2020-11-28

**Authors:** Yoann Maitre, Pierre Micheneau, Alexis Delpierre, Rachid Mahalli, Marie Guerin, Gilles Amador, Frederic Denis

**Affiliations:** 1Emergency Department, Montpellier University Hospital, 34259 Montpellier, France; maitreyoann@yahoo.fr; 2EA 2415, Aide à la Décision pour une Médecine Personnalisée, Université de Montpellier, 34090 Montpellier, France; 3Department of Odontology, Tours University Hospital, 37261 Tours, France; pierre.micheneau@etu.uca.fr (P.M.); alexis.delpierre@etu.uca.fr (A.D.); rachid.mahalli@etu.uca.fr (R.M.); 4Faculty of Dentistry, Clermont-Ferrand University, 63000 Clermont-Ferrand, France; marie.guerin@etu.uca.fr; 5Faculty of Dentistry, Nantes University, 44035 Nantes, France; gilles.amadordelvalle@chu-nantes.fr; 6EA 75-05 Education, Ethique, Santé, Faculté de Médecine, Université François-Rabelais, 37000 Tours, France

**Keywords:** mental health, oral microbiota, oral microbiome, mental disorders

## Abstract

This systematic review aims to investigate the role of the oral microbiome in the pathophysiology of mental health disorders and to appraise the methodological quality of research of the oral–brain axis which is a growing interest area. The PRISMA guideline was adopted, to carry out an electronic search through the MEDLINE database, to identify studies that have explored the role of the oral microbiome in the pathophysiology of mental health disorders published from 2000 up to June 2020. The search resulted in 140 records; after exclusions, a total of 22 papers were included in the present review. In accordance with the role of the oral microbiome in the pathophysiology of mental disorders, four mental disorders were identified: Alzheimer’s disease, dementia, and cognitive disorders; autism spectrum disorder; Down’s syndrome and mental retardation; and Bipolar disorders. Studies argue for correlations between oral microbiota and Alzheimer’s disease, autism spectrum disorders, Down’s syndrome, and bipolar disorders. This field is still under-studied, and studies are needed to clarify the biological links and interconnections between the oral microbiota and the pathophysiology of all mental health disorders. Researchers should focus their efforts to develop research on the oral–brain axis in the future.

## 1. Introduction

While the microbiota defines the population of microbes in a specific ecosystem, the microbiome refers to microorganisms and their genes. These definitions are important because bacteria that live on different parts of the body (skin, gut, and oral cavity) prefer different nutrients and perform different functions. There is an oral microbiota of the mouth, a microbiota of the skin that has many subcategories (e.g., the armpits, nose, feet), and gut microbiota, among many others. The human microbiota is increasingly recognized as a superorganism that is believed to play a role in health and disease [1]. The number of microorganisms inhabiting the human body has been estimated at about 10^14^ prokaryotic organisms with an absolute number of species varying according to the microbiome’s body location [2]. As a rough estimate, the 1000 bacterial species in the gut have 2000 genes per species, which yields 2,000,000 genes, 100 times the approximately 20,000 human genes [3]. This includes bacteria, fungi, viruses, protozoa, and archaea, whose diversity will vary from person to person depending on lifestyle and physiological differences [4]. Disruptions to these different microbiomes are increasingly becoming associated with numerous inflammatory, immune, and nervous system-related diseases by a communication pathway called the microbiome–brain axis [5,6,7,8,9,10].

Research supports microbe-brain interactions, most notably with anxiety and depressive-like behaviors, with accumulating evidence pointing to specific microbial genes that can regulate neurotransmitter activity [11]. The gut microbiome has been shown to play a major role in the development and function of the hypothalamic–pituitary–adrenal axis, [12] which mediates the stress response and is of interest in a range of psychiatric disorders, in particular depression and anxiety disorders. Our gut bacteria also heavily influence the immune system [13] and may represent the link to the immune dysfunction that is characteristic of mental illnesses such as depression and schizophrenia. In mice, *Clostridium* produces dopamine. This neurotransmitter enables communication within the nervous system and is a molecule that directly influences behavior [14]. A strong correlation was shown between functional disorders caused by dysbiosis and the subsequent presence of mood disorders [15]. The dysregulation of the gut–brain axis has emerged as a possible area of research to better understand the pathophysiology of mental disorders and has provided psychiatry with a new paradigm from which to approach mental illness [16,17,18,19].

In the same way, the human oral microbiota has become a new research focus area aimed at promoting the progress of disease diagnosis, complimenting disease treatment, and developing personalised medicines. Evidence suggests that disturbance to the oral microbiota ecological balance can cause a series of infectious oral diseases including dental caries and periodontal diseases. The oral microbiota is also associated with several systemic diseases, including but not limited to cardiovascular disease, pneumonia, rheumatoid arthritis, pancreatic cancer, colorectal cancer, oesophageal cancer, stroke, and adverse pregnancy outcomes. Accordingly, the oral microbiota is increasingly being recognized as a potential biomarker for human diseases [20,21,22,23]. The oral microbiota is composed of a large number of microorganisms housed in a complex environment, that encompasses distinct, small microbial habitats, such as teeth, buccal mucosa, soft and hard palate, and tongue, which form a species-rich heterogeneous ecological system [24]. About 50 to 100 billion bacteria have been identified in the oral cavity and 600 prevalent taxa at the species level, with distinct subsets predominating different habitats [25]. These species belong to 185 genera and 12 phyla, of which approximately 54% are officially named, 14% are unnamed (but cultivated) and 32% are known only as uncultivated phylotypes [26]. The 12 phyla are Firmicutes, Fusobacteria, Proteobacteria, Actinobacteria, Bacteroidetes, Chlamydiae, Chloroflexi, Spirochaetes, Synergistetes, Saccharibacteria and Gracilibacteria [27]. Oral-derived bacteria such as *Porphyromonas gingivalis* and *Actinobacillus actinomycetemcomitans* can colonize the intestines and persist there, leading to activation of the intestinal immune system and chronic inflammation [28].

Oral microbiota such as *Streptococcus mutans*, *Porphyromonas gingivalis*, and *Gemella haemolysans* may play a role in cardiovascular disease [29] and oral pathogens, especially *Porphyromonas gingivalis* and *Aggregatibacter actinomycetemcomitans* are associated with a high risk of pancreatic cancer [30]. In general, the oral microbiota has an impact on the health of the body by digesting food. Salivary DNA sequencing has shown that core species abundance ratios significantly correlate with diet. For example, the abundance of *Neisseria* and *Haemophilus* differs between the hunter–gatherers population living in the western part of the island of Luzon, in the east of this island and in the central part of the island of Palawan (Philippines) [31]. The oral microbiota can also produce metabolites in the mouth that can affect the development of a range of oral diseases. Microorganisms found on the surfaces of teeth tend to form multispecies biofilm communities that are often embedded in a matrix of extracellular polymeric substances (EPS). The production of EPS and acidic metabolites are closely related to the oral microbiota for dental caries [32]. In periodontal diseases, polymicrobial communities induce a dysregulated and destructive host response through a mechanism referred to as polymicrobial synergy and dysbiosis induced by a pathogenic triad of *Porphyromonas gingivalis*, *Tannerella forsythia* and *Treponema denticola* named the red complex [33]. The oral microbiota directly influences dental caries and periodontal diseases [34]. Specific genetic composition and structure of the microbial community can even be identified in the pathogenesis and evolution of periodontal disease. When the microbiota isolated from patients with periodontal disease were compared with health controls, the most striking difference was the relative proportions of the four most abundant phyla, Bacteroidetes, Actinobacteria, Proteobacteria, and Firmicutes [35,36]. Likewise, we observed change in oral microbiota in systemic diseases like, for example, an elevation of proinflammatory cytokines in saliva patients suffer from oral squamous cell carcinoma [37]. A link between composition and diversity of the oral microbiota and type II diabetes is suspected [38]. There are indications that periodontitis precedes rheumatoid arthritis [39] and saliva would be able to provide informative clinical markers to obesity [40]. Oral bacteria could, therefore, be linked to or serve as biomarkers for certain systemic diseases. However, it remains to be established whether there is a causal relationship between the oral microbiome and these systemic disorders.

The oral microbiome, therefore, appears crucial for health as it can cause both oral and systemic diseases. Identifying the microbiome in health is the first step of human microbiome research, after which it is necessary to understand the role of the microbiome in the alteration of functional and metabolic pathways associated with the diseased states.

In this perspective, the gut–brain axis is gaining ever more traction in fields investigating the biological and physiological basis of psychiatric, neurodevelopmental, and neurodegenerative and age-related disorders. On the other hand, the impact of the oral microbiota and the physiopathological mechanisms that it generates, to our knowledge, are little emphasized in their relationship with mental disorders.

Therefore, the aim of this systematic review was to investigate the role of the oral microbiome in the pathophysiology of mental health disorders, to appraise the methodological quality of the oral–brain axis research, a young yet fast growing field of interest.

## 2. Materials and Methods

The present systematic review was conducted according to the PRISMA guidelines for Systematic Reviews [41].

### 2.1. Search Strategy

An electronic search was conducted through the MEDLINE (PubMed) database to identify publications that met the inclusion criteria. High-throughput sequencing techniques for genetic material developed in the 2000s have led to a better understanding of the nature of host–microbiota interactions, the interactions of microorganisms with each other, and their impact on health [42]. That is why the search was performed from 2000 up to June 2020, in order to identify the studies that explore the role of the oral microbiome in the pathophysiology of mental health disorders, using the following search terms and keywords alone or in combination with the Boolean operator “AND”/ “OR” according to the following equation [Mouth/microbiology [Mesh]) OR “Mouth Diseases/microbiology”[Mesh]) OR “Saliva/microbiology”[Mesh:NoExp]) OR “Mouth Diseases/immunology”[Mesh]) AND “Mental Disorders”[Mesh]].

### 2.2. Study Detection

References of the eligible studies on the topic were manually checked, and two independent operators (F.D. and Y.M.) screened the studies according to the inclusion/exclusion criteria. 

#### Inclusion and Exclusion Criteria

We included experimental or clinical studies (longitudinal, cross-sectional, or randomised studies) that (1) explored the link between the oral microbiome and mental disorders from the inflammatory perspective or (2) explored an association between oral microbiota dysbiosis and mental health symptoms. We excluded conferences, abstracts, reviews, and editorials. Broadly defined, dysbiosis is any change in the composition of resident commensal communities relative to the community found in healthy individuals. As dysbiosis in the microbiome is associated with many diseases [43,44,45,46,47], changes observed in the oral microbial composition are hypothesised to contribute to the onset and/or persistence of mental disorders.

### 2.3. Data Collection

Two reviewers (F.D. and Y.M.) independently screened the list of titles and abstracts to identify the potentially relevant papers based on the inclusion criteria. If the abstracts were judged to contain insufficient information, then the full studies were reviewed to decide whether they should be included based on the selection criteria. A scan of the references of the previously selected articles completed the selection. When a discrepancy in the selection decision occurred, the two reviewers engaged in discussion until a consensus was reached. If needed, a third reviewer (G.A.) resolved the possible conflicts concerning eligibility.

## 3. Results

### 3.1. Study Selection

The initial studies retrieved from the databases were first selected, and studies that met the eligibility criteria were reviewed and analysed. From 140 studies, 19 were selected. The complementary detection resulted in the selection of three additional articles. Taken together, a total of 22 articles were included in the analysis (Figure 1).

Each study that met the inclusion criteria was analysed from several aspects, such as authors, date of publication, the time when the study was conducted, study design, objectives, results, and limitations.

Of the 22 studies included in the analysis (Table 1), four distinct themes in mental health disorders emerged: Alzheimer’s disease (AD), dementia, and cognitive disorders [48,49,50,51,52,53,54,55,56,57,58,59,60]; autism spectrum disorder (ASD) [61,62,63]; Down’s syndrome (DS) and mental retardation [64,65,66,67,68]; and bipolar disorders [69].

### 3.2. Alzheimer’s Disease, Dementia, and Cognitive Disorders

Most mental health disorders are related to dementia. Dementia describes a group of symptoms associated with cognitive disorders, such as a decline in memory, reasoning, or other thinking skills. Many different types of dementia exist, and many conditions cause it. AD is the most common cause of dementia [70]. Although it is a neurodegenerative disease, AD and its symptoms are outlined in the Diagnostic and Statistical Manual of Mental Disorders fifth edition (DMS V) and in the International Classifications of Diseases, 10th Revision (ICD-10) and is recognized as a mental disorder with a neurological origin. In cross-sectional studies, many authors have evoked a correlation between the presence of an inflammatory process whose origin could be periodontal and the presence of dementia or AD [48,49,50]. The main markers of inflammation to be studied were the cytokine levels produced by unstimulated and LPS-stimulated peripheral blood leukocytes [50] or the plasma CRP [49]. Studies using an animal model of periodontitis and human post-mortem brain tissues from subjects with AD strongly suggest that a Gram-negative periodontal pathogen, *Porphyromonas gingivalis* and/or its product gingipain is/are translocated to the brain [51,52]. Taguchi et al. [53] showed that an increase in periodontal pocket depth tended to be associated with the number of lacunar infarctions in Japanese adults of both sexes. Kamer et al. [54] showed that both the plasma TNF-α level and the number of positive tests for antibodies against periodontal bacteria were elevated in AD and independently associated with AD, suggesting that antibody levels to periodontal bacteria associate with AD and might help improve the clinical diagnosis of AD. In this perspective, serological marker like gingivalis IgG of periodontitis could be associated with impaired delayed memory and calculation [55,56]. Dominy et al. [57] suggested that *Porphyromonas gingivalis* and gingipains in the brain play a central role in the pathogenesis of AD, providing a new conceptual framework for disease treatment and designed an orally bioavailable, brain-penetrant Kgp inhibitor currently being tested in human clinical studies for AD. Kgp is a component of a protease essential for the pathogenicity of *Porphyromonas gingivalis* (gingipain) found in the cerebral cortex. However, even with a Mendelian randomization approach that takes into account the genetic variability of periodontal pathogens to investigate the causal relationship between periodontitis and AD bidirectionally in the population of European ancestry, Sun et al. [58] did not find convincing evidence to support periodontitis being a causal factor for the development of AD.

In mice, Watanabe et al. showed that the chronic presence of cnm-positive *Streptococcus mutans* hematogenously induces cerebral haemorrhage by disrupting the blood–brain barrier. The collagen-binding activity of cnm-positive *Streptococcus mutans* is closely related to the occurrence of deep cerebral microbleeds and might be a risk factor for cognitive impairment [59]. 

Patients with hepatic encephalopathy, which is associated with cognitive dysfunction, have unique microbial signatures in their stool and salivary microbiota [60].

Lastly, using molecular and immunological techniques, Riviere et al. observed Treponema in frozen trigeminal ganglia, brainstem, and cortex samples from human subjects [48]

### 3.3. Autism Spectrum Disorder

ASD is a complex neurological and developmental disorder whose aetiology likely involves an interplay between genetic and environmental factors, as well as systemic inflammations [71].

In a cross-sectional study, a comparative analysis of oral microbiota in ASD children by a sequencing method showed an alteration in the composition and taxonomy of the oral microbiota in ASD patients compared to healthy controls. In this case, dysbiosis in the oral microbial community of ASD patients suggested a potential role for microorganisms in the progression of this disease [61]. Hicks et al. [62] identified distinct oral micro transcriptomic activity in ASD children and suggests oral microbiome profiling as a potential tool to evaluate ASD status. Kong et al. [63], in a cross-sectional pilot study, conducted a comparative analysis of the oral and gut microbiomes in patients with ASD. This study identified distinct features of the gut and salivary microbiota that differ between individuals with and without an ASD diagnosis.

### 3.4. Down’s Syndrome and Mental Retardation

DS is a genetic disorder caused by the presence of all or part of a third copy of chromosome 21 and is associated with mild to moderate intellectual disability and facial characteristics.

Studies showed [64,65,66] that the DS population developed an earlier and more extensive periodontal breakdown. However, Reuland-Bosma et al. [65] report an absence of specific flora in adults with DS compared to a control group. 

Despite this finding, several studies have identified certain specificities in DS patient’s microbiota. With similar levels of gingival inflammation and plaque accumulation with Porphyromonas gingivalis with specific genotype (type II fimA) may be a determinant for DS early-onset periodontitis as well as non-DS populations [64]. The score of Streptococcus *mutans* in saliva close to type k was higher in DS and mental retardation than in control [67]. Khocht et al. [68] have reported that the subgingival microbiological profiles of adults with DS with periodontitis are similar to those found in patients with chronic periodontitis. These authors note, however, significant differences between the groups for some bacterial species (*Actinobacillus actinomycetemcomitans*, *Porphyromonas gingivalis*, *Tannerella forsythia*, *Prevotella intermedia*, *Selenomonas noxia*, *Propionibacterium acnes gordonii*, *Streptococcus mitis*, *and Streptococcus oralis*, *Streptococcus constellatus Treponema Socranskii*). These differences highlight that patients with DS have higher levels of certain bacterial species and specific associations between certain bacterial species and the loss of periodontal attachment. However, the majority of bacteria identified are opportunistic bacteria suggesting the role of a carried by hands contamination (thumb sucking, finger biting). The association between IL-1 (which is a proinflammatory marker of periodontal diseases) polymorphic genotypes and periodontitis differ between the DS and non-DS subjects [66].

### 3.5. Bipolar Disorders

Bipolar disorder is a mental health condition that causes extreme mood swings that include emotional highs (mania or hypomania) and lows (depression).

Cunha et al. [69] showed that Bipolar disorder was associated with an increased risk for periodontitis and a higher frequency of all evaluated periodontopathogens, as well as a higher total bacterial load like *Aggregatibacter actinomycetemcomitans* and *Porphyromonas gingivalis.* The prevalence of periodontitis and the increase of the bacterial load concerning *Aggregatibacter actinomycetemcomitans* and *Porphyromonas gingivalis* is especially associated with the depressive phases compared to the manic or euthymic phases [69].

## 4. Discussion

While many studies highlight the link between the gut microbiota and mental disorders, studies on the impact of the oral microbiota and the pathophysiological mechanisms are little emphasized in their relationship with mental disorders. We identified 22 studies on the subject only one of which links bipolar disorders to the oral microbiota. No studies have been published on the link between the oral microbiota and schizophrenia or depressive disorders despite these being major psychiatric disorders. The studies carried out in this area mainly concern cognitive disorders associated with AD (13/22 identified studies). The public health problem linked to the growth of AD, dementia, and cognitive disorders in the general population due to its overall aging [72] is certainly a cause that might explain this preferential research orientation.

The identification of a specific microflora in patients with mental disorders suggests that these disorders could influence the oral microbiome. This hypothesis presumes that the CNS (Central nervous System) is capable of modifying the oral environment to favour the preferential selection of particular microbes, as is the case for the intestinal flora [73]. This hypothesis is tempered by the poor oral condition usually associated with patients with mental disorders. Among persons with severe mental illness, dental caries and periodontal measurement indexes often reach twice the level found in the general population [74,75]. Many combined factors contribute to the poor oral health of these individuals, including orals infectious diseases interacting with the metabolic disturbances induced by antipsychotic treatments (e.g., diabetes, obesity), and also poor diet and lifestyle choices (e.g., high sugar diet, use of psychoactive substances such as tobacco, and inadequate oral hygiene) [76,77].

Periodontal diseases and dental caries are the two main oral pathologies in humans [78]. The increase in oral bacteria load (*Porphyromonas gingivalis*, *Streptococcus mutans*, *Treponema denticola*, etc.), and the elevated concentrations of the inflammatory markers they induce (TNF α and cytokines), appear to be found in patients suffering from mental illness. Many psychiatric conditions including AD are associated with chronic low-grade inflammation and elevated level of pro-inflammatory cytokines [79]. Gut microbiota disturbances may represent a possible mechanism linking chronic stress, cytokine production and neuropsychiatric disorders such as depression [80]. Oral microbiome disturbances may represent another such link achieved through the translocation of microbes from the oral cavity into the bloodstream. The oral microbiome and the gastrointestinal microbiomes share many common microbes, another potential link through which the two sites both contribute to inflammatory diseases. Taken together, it is not surprising that oral and intestinal microbiota show concordant disease associations [81]. The oral dysbiosis and the inflammatory phenomena it induces could play a part in the development of mental illnesses, just like intestinal dysbiosis [82]. Since the oral cavity represents the main entry point to the intestinal tract, it can be assumed that the accumulated evidence regarding the link between the intestinal flora and mental disorder is in favour of future studies related to the oral microbiota [26].

Mental illnesses (e.g., schizophrenia, depressive syndromes) involve neurotransmitter dysfunction. Gut bacteria can directly produce neurotransmitters used in the human body including GABA, serotonin, noradrenaline, acetylcholine and dopamine [83]. However, the quantities produced by bacteria are relatively small and unlikely to influence human neurotransmission directly to any great extent. The oral microbiota can also produce metabolites. In 2019, Lin et al. explored the association between fluctuations within the oral microbiome and the brain functional network in smokers and suggested the biological pathways underlying for the microbiome–brain link, but were unable to assign causality among the features [84]. 

In the interdental biofilm of young and healthy periodontal subjects with no signs of periodontal disease, there are periodontopathogen bacteria such as *Porphyromonas gingivalis* [85] that also appear to be involved in the development of Alzheimer’s disease. Thus, the setting up as the youngest age of individual oral prophylaxis directed towards the interdental spaces, with the use of interdental brushes, would help to combat dysbiosis of the oral microbiota and could potentially prevent the onset of diseases such as Alzheimer’s disease [34,86].

Differences in the gut microbiome profiles of people with ASD have been reported like an increase in the abundance of *Clostridium* species [87] and high levels of *Sutterella* [88], although the results vary from study to study. In addition, the oral microbiome of children with ASD differs from that of neurotypical children in several taxa predominantly related to energy metabolism and lysine degradation pathways [62]. Although several studies to investigate the effects of various probiotics on ASD symptoms, they are greatly limited by small sample sizes and methodological challenges, and, as such, it is difficult to draw any concrete conclusions [89]. Concerning the oral microbiota in people with ASD, multiple studies have identified species specifically associated with this disorder [61,62,63].

In accordance with reports of higher level of certain bacterial species in the oral microbiota of patients with, DS, Biagi et al. [90] highlight that the general makeup of their microbiomes were similar to those of healthy individuals, though, some subpopulations of bacteria were more or less abundant in patients with DS. Individuals with DS have elevated levels of *Parasporobacterium* and *Sutterella* and reduced levels of *Veillonellaceae*. Specifically, in this study, *Sutterella* abundance positively correlated with the Aberrant Behaviour Checklist (ABC) scores individuals with DS. This positive correlation indicates that in the gut at least, this bacterium may contribute to the maladaptive behaviour in patients with various conditions. 

Several studies examine the composition of the microbiome in patients with bipolar disorder [91] and report different levels of *Faecalibacterium*. In 2019, Coello et al. reported that *Flavonifractor*, a bacterial genus that can induce oxidative stress and inflammation, was associated with bipolar disorder and that probiotics from *Lactobacillus* and *Bifidobacterium* appear to have therapeutic potential [92]. Cunha et al. [69] who, to the best of our knowledge, are the only authors to have studied the oral microbiota composition in patients with bipolar disorder, found a higher frequency of all periodontopathogens in this populations.

Despite the mounting evidence in support of a link between the oral microbiome and mental health, it must be kept in mind that the relative abundance and diversity of bacteria vary considerably from one individual to another [93]. As a consequence, studies generally only focus on simple correlations between the abundance of particular taxa and the pathological conditions associated with the host. These association patterns do not allow a discrimination between effect and cause [94].

A consensus has yet to be reached on a precise definition for the characteristics of a bidirectional relationship between the oral microbiome and the brain as it can be advanced in the context of the gut–brain axis [95]. The identification of specific oral dysbiosis signatures associated with various mental disorders raises the question of whether the CNS modulate the oral microbiome. This would constitute the complementary element of a bidirectional relationship suggesting the existence of an oral cerebral axis equivalent to the gastrointestinal axis. However, if, within the framework of the intestinal–brain axis, various modulation mechanisms are well established (e.g., alteration of mucus and biofilm production, motility, permeability and immune function [95], no mechanism underlying the particular microbial signature in patients with mental disorders has been studied or even envisaged in the many studies included in this review.

## 5. Limitations

Although the studies in our review argue for an interrelation between oral and mental pathologies, they should not be over-interpreted due to numerous limitations. The studies are based on relatively small samples and cross-sectional surveys that allow only hypotheses about the existence of an oral–brain axis. The small size of these surveys prevented the subdivision of participants into phenotypic subtypes of their mental disorders. Schizophrenia and mood disorders were not studied. The studies were not designed to take into account the progressive nature of oral dysbiosis and mental disorders over time. Thus, most of the findings still appear to be preliminary, as generally acknowledged by their authors.

The difficulties of diagnosis of psychiatric disorders [96] and the absence of consensus in the definition of the principal oral dysbiosis [97] have led to heterogeneity of the patients’ populations between the various studies. In the same way, the methodological variations in the collection of the bacterial samples and the targeted species induce heterogeneity of the results. In this context, it is difficult to have a precise idea of the role of the oral microbiome in the pathophysiology of mental health disorders.

Finally, the evidence for a two-way causal link between the oral microbiota and mental disorders is limited in humans due to a lack of analysis of confounding factors. While age and gender are sometimes considered, important parameters such as lifestyle habits (e.g., smoking, diet, oral hygiene) or functional or immune disorders related to mental disorders or somatic pathologies are rarely taken into account in the analysis.

## 6. Perspectives

To clarify the nature of the link between the oral microbiota and the CNS, it appears essential to develop and validate a methodological approach based on a clear definition of oral dysbiosis and the mental disorders studied.

Only follow-up over time will shed light on the link between the oral microbiome and the various psychiatric disorders and will make it possible to take into account their evolution over the life of the patients. To confirm the existence of an oral–brain axis, a prospective study in a cohort of sufficient size should evaluate the role of the different confounding factors reflecting the entanglement between mental disorders and oral dysbiosis (level of oral hygiene, oral monitoring, various medications).

As suggested by the work on the gut–brain axis, it can be assumed that the identification of a particular microbiome associated with a specific mental disorder might eventually constitute an objective complementary diagnostic element of the disease. In the same way, if the bidirectional link between the oral microbiome and the CNS were confirmed, a rebalancing of microbial homeostasis through the use of probiotics could represent a potential therapeutic axis. 

## 7. Conclusions

Our study argues for correlations between oral microbiota and AD, dementia or cognitive disorders, ASDs, DS, mental retardation, and bipolar disorders. This field remains little explored, and further studies are needed with a larger sample size involving different type of mental disease are required to clarify the biological link or interconnections between the oral microbiota and all mental health disorders and to develop consistent patterns to generate more concrete data.

The concept of the oral brain axis is an area with growing interest that is likely to facilitate the diagnosis of mental disorders by providing novel disease-specific biomarkers. Future research efforts should focus on developing new therapeutic approaches for better patient management, even though there is still a lot to learn about the oral microbiome, and its potential reaction to therapeutic manipulation. While the literature represents a basis for a reciprocal interaction between the oral microbiome and the brain in the context of mental disorders, the exploratory nature of the studies and their many limitations underline the importance of supporting research in this field.

## Figures and Tables

**Figure 1 jcm-09-03876-f001:**
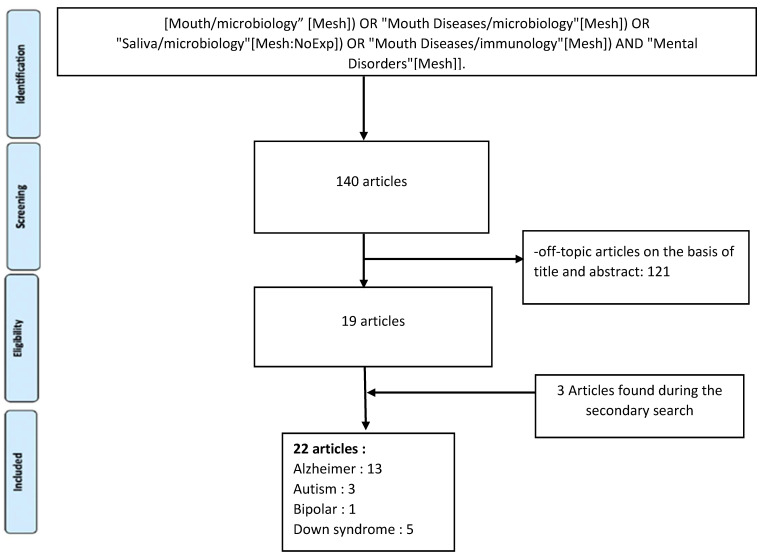
Flow chart of included studies.

**Table 1 jcm-09-03876-t001:** Summary of studies that have examined the relationship between oral microbiota and mental disorders.

Reference(s)	Study Design	Mental Disorder	Objectives	Involved Species	Mains Results and Limitations (*)
Amano et al., 2001 [64] (*n* = 67)	Cross sectional	Down syndrome	Comparison of bacterial prevalence in samples of subgingival plaque and detection of *Pg* fimA genotypes in DS patients and non-DS subjects with mental disabilities.	*A. actinomycetemcomitans*, *P. gingivalis*, *B. forsythus*, *T. denticola*, *P. intermedia*, *P. nigrescens*, *C. ochracea*, *C. sputigena*, *C. rectus*, *E. corrodens*	Early-onset periodontitis in DS is mainly due to the more susceptible host for the causative microbial agents including *P. gingivalis* with type II fimA. * Need to develop genomic analyses to better detect allelic modulations and specific bacterial polymorphisms involved in the development of early periodontitis in DS patients.
Reuland-Bosma et al., 2001 [65] (*n* = 17/17)	Cross sectional	Down syndrome	Investigation of the subgingival microflora in DS subjects and other mentally retarded individuals. Investigation of the subgingival microflora of a ‘‘low-risk’’ and a “high-risk’’ group formed in DS subjects.	*A. actinomycetemcomitans*, *P. gingivalis*, *P. intermedia*, *B. forsythus*, *P. micros*, *F. nucleatum*, *C. rectus*	The higher prevalence of periodontal disease in DS subjects is probably related to the impaired host-response and not to the occurrence of specific periodontal pathogens. * Insufficient consideration of patient age.
Riviere et al., 2002 [48] (*n* = 34)	Cross sectional	Alzheimer’s disease Cognitive disorders Dementia	Assessment of the oral treponema presence in the human brain, the hippocampus and the trigeminal ganglia in AD patients and controls.	*T. amylovorum*, *T. denticola*, *T. maltophilum*, *T. medium T. pectinovorum*, *T. socranskii*, *T. vincentii*	Presence of oral *Treponema* in the trigeminal ganglia, the brainstem, and the cortex of human subjects with AD. AD patients were more likely to have Treponema than controls, and they had more different Treponema species in brain than controls. * Postmortem contamination of tissues or assays cannot be excluded
Linossier et al., 2008 [67] (*n* = 59/60/60)	Cross sectional	Down syndrome and Mental retardation	Comparison of the concentration and serotype of *Sm* in the saliva of people with DS, mentally retarded people and healthy controls	*S. mutans*	The score of *Sm* in saliva was higher in DS and mental retarded than in controls. * The deficient hygiene in DS and mental retardation could lead to a wider colonization of the biofilm by these bacteria
Kamer et al., 2009 [54] (*n* = 18/16)	Case-control	Alzheimer’s disease Cognitive disorders Dementia	Test the hypothesis that concentration of TNF α and antibodies raised to periodontal bacteria would be more elevated in AD compared to normal controls. Investigation of TNF-alpha and antibodies as diagnostic biomarkers for AD.	*A. actinomycetemcomitans*, *T. forsythia*, *P. gingivalis*	AD patients have both elevated cytokine expression and antibodies to periodontal bacteria. These measures could contribute to the diagnosis of AD. * Small sample size and study design
Noble et al., 2009 [56] (*n* = 2355)	Cross sectional	Alzheimer’s disease Cognitive disorders Dementia	Assessment of the relationship between systemic exposure to periodontal pathogens and cognitive test outcomes.	*P gingivalis*	Association between a serological marker of a common periodontitis pathogen (*Pg* immunoglobulin G) and poor cognitive test performance. * Difficulty in assessing cognition and the impact of socioeconomic status throughout life on the prevalence of periodontal disease.
Khocht et al., 2011 [66] (*n* = 44/66/83)	Cross sectional	Down syndrome	Investigation of the distribution of IL-1 genotypes’ in a DS population and examine the association of IL-1 polymorphism with their periodontal status.	Not relevant	Association between the carriage of the IL-1 rare alleles and periodontitis differed between Down syndrome and healthy patients. DS individuals are more susceptible to periodontal loss of clinical attachment than non-Down individuals. * Lack of evaluation of the immune characteristics of patients.
Rai et al., 2012 [49] (*n* = 55/32)	Cross sectional	Alzheimer’s disease Cognitive disorders Dementia	Evaluation of the role of inflammatory mediators between periodontitis and dementia.	Not relevant	Possible role of inflammatory mediators between periodontitis and dementia. * Hospital patient selection, study design, small sample size.
Khocht et al., 2012 [68] (*n* = 212)	Cross sectional	Down syndrome	Determine the subgingival microbial profile in adult-aged individuals with DS. Investigation of the subgingival microbial profile of Down syndrome adults matched with non-DS mentally retarded subjects and mentally normal subjects. Assessment of the association between bacterial species and loss of clinical periodontal attachment in individuals with and without DS.	*A. gerencseriae*, *A. israelii*, *A. naeslundii*, *A. naeslundii*, *A. odontolyticus*, *V. parvula*, *S. gordonii*, *S. intermedius*, *S. mitis*, *S. oralis*, *S. sanguinis*, *A. actinomycetemcomitans*, *C. gingivalis*, *C. ochracea*, *C. sputigena*, *E. corrodens*, *C. gracilis*, *C. rectus*, *C. showae*, *E. nodatum*, *F. nucleatum ss nucleatum F. nucleatum ss polymorphum*, *F. nucleatum ss vincentii*, *F. periodonticum*, *P. micra*, *P. intermedia*, *P. nigrescens*, *S. constellatus*, *T. forsythia P. gingivalis*, *T. denticola*, *E. saburreum*, *G. morbillorum*, *L. buccalis*, *N. mucosa*, *P. acnes*, *P. melaninogenica*, *S. Anginosus*, *S. noxia*, *T. socranskii*	Similarities between the subgingival microbial composition of individuals with and without DS. Individuals with DS show higher levels of some bacterial species and specific associations between certain bacterial species and loss of periodontal attachment. * Lack of evaluation of the impact of the specific immune status of DS patients.
Poole et al., 2013 [52] (*n* = 10)	Cross sectional	Alzheimer’s disease Cognitive disorders dementia	Investigation of the link between periodontal disease and AD with a view to identifying the major periodontal disease bacteria and/or bacterial components in brain tissue from 12 h post-mortem delay.	*T. denticola*, *T. forsythia*, *P. gingivalis*	Lipopolysaccharide from *Pg* can access the AD brain during life suggested an inflammatory role in the existing AD pathology. * Small sample size.
Taguchi et al., 2013 [53] (*n* = 110)	Cross sectional	Alzheimer’s disease Cognitive disorders Dementia	Evaluation of the associations between tooth loss, chronic oral inflammation and asymptomatic lacunar infarction.	Not relevant	Periodontal disease progression tended to be associated with an increased number of lacunar infarctions. * Hospital patient selection, study design, small sample size, uncertain reproducibility of measure of periodontal defects intra- and interexaminer, and lack of consideration of other risk factors associated with deficient infarction (hypertension and high total plasma homocysteine).
Noble et al., 2014 [55] (*n* = 110/109)	A case-cohort	Alzheimer’s disease Cognitive disorders Dementia	Evaluation of the serum IgG directed against periodontal microbiota as possible predictors of incident AD.	*P. gingivalis*, *T. forsythia*, *A. actinomycetemcomitans Y4*, *T. denticola*, *C. rectus*, *E. nodatum*, *A. naeslundii genospecies-2*	Serum IgG high concentrations to common periodontal microbiota represents a risk factor for developing incident AD. * Small sample size.
Watanabe et al., 2016 [59] (*n* = 279)	Cross sectional	Alzheimer’s disease Cognitive disorders Dementia	Evaluation of the impact of *Sm* expression of the Cnm protein on the localization of cerebral microbleeds and its consequences on cognitive decline.	*S. Mutans*	The collagen binding activity of cnm-positive *Sm* was a risk factor to the occurrence of deep cerebral microbleeds and may be a risk factor for cognitive impairment. * Small sample size, study design, institutional recruitment and limited evaluation of cognitive deficit.
Sochocka et al., 2017 [50]	Cross sectional	Alzheimer’s disease Cognitive disorders	Evaluation of the correlation between periodontal health	Not relevant	The presence of cognitive decline and the additional source of proinflammatory
(*n* = 128)		Dementia	status and cognitive abilities with relative changes in systemic measures of pro-inflammatory and anti-inflammatory cytokines as a reflection of systemic inflammation.		mediators, like periodontal health problems, aggravate the systemic inflammation and neurodegenerative lesions. * Biases associated with the statistical definition of level of inflammatory state.
Hicks et al., 2018 [62] (*n* = 180/106/60	Cross sectional	Autism spectrum disorder	Evaluation of oral microbial taxa in children with ASD, nonautistic developmental delay and typically developing.	Oral bacterial taxa	Identification of distinct oral microtranscriptomic activity in ASD children relative to both typically developing controls peers and nonautistic peers with developmental delay. * Failure to take into account confounding factors (gastrointestinal disorders, social behaviour).
Ilievski et al., 2018 [51] (*n* = 10/10)	longitudinal with mice.	Alzheimer’s disease	Test the hypothesis that repeated exposure of wild type C57BL/6 mice to orally administered *Pg* results in neuroinflammation, neurodegeneration, microgliosis, astrogliosis and formation of intra- and extracellular amyloid plaque and neurofibrillary tangles.	*P. gingivalis*	*Pg*/gingipain was detected in the hippocampi of mice exposed Neuropathological features of AD have been observed in young adult mice after repeated oral application of *Pg* * No direct or indirect mechanism that could explain these changes could be identified.
Qiao et al., 2018 [61] (*n* = 32/27)	Cross sectional	Autism spectrum disorder	Exploration of the differences in the salivary and dental microbiota between children with ASD and healthy control.	Oral bacterial taxa	Identification of oral microbial taxa associated with ASD. Identification of characteristic differences in salivary and dental microbiota between people with ASDs and healthy people. * Small sample size, study design.
Bajaj et al., 2019 [60] (*n* = 279)	Cross sectional	Alzheimer’s disease Cognitive disorders Dementia	Determination of the gut and salivary microbial profiles of patients with and without MHE.	Oral bacterial taxa	A specific microbial signature in the stool and salivary microbiota is associated with individual cognitive impairment in patients with MHE due to cirrhosis. * Study design, small sample size, Non-differentiation of initial or recurrent cases of MHE.
Cunha et al., 2019 [69] (*n* = 176/176)	Case-control	Bipolar disorders	Evaluation of the epidemiological and microbiological aspects of the potential association between bipolar affective disorder and periodontitis.	*A. actinomycetemcomitans*, *T. denticola*, *P. gingivalis*	Individuals with bipolar affective disorder presented higher frequency of *Aa* and *Pg*. * Small sample size, Study design preventing the detection of any temporal influence among bipolar disorders, periodontal condition and microbiological findings.
Dominy et al., 2019 [57]	Experimental study conducted with mice	Alzheimer’s disease Cognitive disorders Dementia	Investigation of the prevalence of *Pg* in the AD brain and to elucidate possible *Pg* dependent mechanisms of action for neurodegeneration and AD pathology.	*P. gingivalis*	*Pg* and gingipains in the brain play a central role in the pathogenesis of AD. Gingipain inhibitors could be valuable for treating *Pg* brain colonization and neurodegeneration in Alzheimer’s disease.
Kong et al., 2019 [63] (*n* = 20/19)	Cross sectional	Autism spectrum disorder	Characterization of oral and intestinal microbiomes of patients with ASD compared to neurotypical patients.	Oral bacterial taxa	Identified distinct features of gut and salivary microbiota that differ between individuals with and without an ASD diagnosis. * Large inter-individual variability induced by the use of both sibling and parental as neurotypical controls, small sample size.
Sun et al., 2020 [58] (*n* = 4924 /7301)	A bidirectional Mendelian randomization study	Alzheimer’s disease Cognitive disorders Dementia	Examination of the potential causal relationship between chronic periodontitis and AD bidirectionally in the population of European ancestry.	Not relevant	No evidence that periodontitis is a causative factor of AD and genetic responsibility for AD on the risk of periodontitis from the analysis of data from the Genome-Wide Association Studies on periodontitis and AD. * Impossibility in the case of Mendelian randomisation to assess the impact of the duration of the periodontal disease on the risk of developing AD and risk of no detection to the causal link between PD and AD induced by the genetic tool used to define PD.

AD: Alzheimer’s disease; ASD: Autism spectrum disorder; DS: Down syndrome; MHE: Minimal hepatic encephalopathy; PD: Periodontal disease; *Pg*: Porphyromonas Gingivalis, *Sm*: Streptococcus mutans; *Aa*: Aggregatibacter actinomycetemcomitans; TNF: Tumor Necrosis Factor; Il: Interleukin. *: Limitations.

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
