# Peer review of "Did the Brain and Oral Microbiota Talk to Each Other? A Review of the Literature"

_jcm, 2020, doi:10.3390/jcm9123876_

Round 1
Reviewer 1 Report
The manuscript reviews the association of the oral microbiota with mental disorders. While the general topic is very interesting, the work is often missing an in-depth investigation of involved species/genera, suggested routes of action and a critical discussion thereof. The causal associations of the routes brain->oral and oral->brain are insufficiently discussed and potential applications or novel insights remain unclear. A deeper discussion of the triangle oral-gut-brain would be helpful. Furthermore, the search terms are quite restrictive, the number of publications for non-AD are quite low and further mental disorders are not included (e.g. schizophrenia). In general the different sections should provide more details and the findings should be more interconnected.
Introduction: a general introduction to the oral microbiome research and its potential is missing.
The focus of mental disorders is underprepresented in the introduction. Provide more emphasize on this topic.
26: rephrase "little explored"
35: nutrients instead of foods
38: role in health
40: provide a range for the number of species
57: 50 to 100 billion: be more precise with this number: is this the total number? how many different species? (see also line 40)
57: check grammar
58: which ones? give examples
59: which ones? give examples
60: which species cause which disease? be more specific
62: which metabolites?
63: how?
64-65: give examples, which degrees linked to which microbial composition
69: check grammar
69: contradictory to previous sentence
2.1: the search terms are very restrictive: "microbiota" and "microbiome" are missing, synonyms for mental disorders and specific diseases should be included.
86: why restricted to 2000?
2.2.1: give examples for dysbiosis
Fig. 1: not clear why the search terms are so restrictive (see above)
Fig. 1: use English expressions for diseases
Fig. 1: what is meant by "out of equation"?
Fig. 1: AD is overrepresented (59.1%), number of publications for other diseases are quite low (1,3,5).
115-116: objectives, results, limitations are missing in Table 1.
3.5. This section is based on only 1 article
Table 1: some interesting information of the references is missing (see comment 115-116), furthermore the involved species in an extra-column and the suggested route of action would add significant value to the overview
Table 1: Sun 2020: what is meant by "not provide convincing evidence" - be more precise. what is claimed in the paper?
Table 1: Poole 2013: Lipopolysaccharide
3.2.1 and 3.2.2 are confusing. Why are they separated? S. mutans is a caries associated species - why is it listed in the perio section?
Discussion should dig deeper into the findings and interconnect them.
Conclusion should provide more details and the potential of the topic.
Author Response
Response to reviewers 1
We thank the Reviewer for giving us an opportunity to substantially improve the content and the presentation of our manuscript. We have modified the article in accordance with your requests. You will find every modification in the text using track changes, and the pages are noted in the answer for every point below.
Reviewers 1
The manuscript reviews the association of the oral microbiota with mental disorders. While the general topic is very interesting, the work is often missing an in-depth investigation of involved species/genera, suggested routes of action and a critical discussion thereof. The causal associations of the routes brain->oral and oral->brain are insufficiently discussed and potential applications or novel insights remain unclear. A deeper discussion of the triangle oral-gut-brain would be helpful. Furthermore, the search terms are quite restrictive, the number of publications for non-AD are quite low and further mental disorders are not included (e.g. schizophrenia). In general, the different sections should provide more details and the findings should be more interconnected.
Response
We thank the reviewer for these suggestions and have made many changes to improve our manuscript.
Introduction:
1/A general introduction to the oral microbiome research and its potential is missing.
Response
As requested, we have introduced research on the oral microbiome and its potential and we have added 2 citations and listed them in the references section, lines 61 to 70.
[20]- Jia, G; Zhi, A; Lai, P.F; Wang, G; Xia, Y; Xiong, Z; Zhang, H; Che, N; Ai, L. The oral microbiota-a mechanistic role for systemic diseases. Br Dent J. 2018, 224, 447–55.
[21]- Sampaio-Maia, B; Caldas, I.M. Pereira M.L. Pérez-Mongiovi, D; Araujo, R. The oral microbiome in health and its implication in oral and systemic diseases. Adv Appl Microbiol. 2016, 97, 171–210.
2/The focus of mental disorders is underprepresented in the introduction. Provide more emphasize on this topic.
Response
As requested, we developed this theme have lines 51 to 56 and we have added 2 citations and listed them in the references section.
[12] Rea, K.; Dinan, T.G.; Cryan, J.F. The microbiome: A key regulator of stress and neuroinflammation. Neurobiol Stress. 2016, 4, 23-33.
[13] Belkaid, Y.; Hand, T.W. Role of the microbiota in immunity and inflammation. Cell. 2014, 157, 121–141.
3/26: rephrase "little explored"
Response
We have changed to "understudied".
4/35: nutrients instead of foods
Response
We have removed foods to nutrients, line 36.
5/38: role in health
Response
We have removed “and”
6/40: provide a range for the number of species
Response
We have added a new sentence lines 42 to 43 and we have added a new citation and listed them in the references section.
[3] Turnbaugh, P.J.; Ley, R.E.; Hamady, M.; Fraser-Liggett, C.M.; Knight, R.; Gordon J.I. The human microbiome project. Nature. 2007, 449, 804-810.
7/57: 50 to 100 billion: be more precise with this number: is this the total number? how many different species? (see also line 40)
Response
We have added a new paragraph lines 74 to 81 and we have added 2 citations and listed them in the references section.
[24]- Zhao, H.; Chu, M.; Huang, Z.; Yang, X.; Ran, S.; Hu, B.; Zhang, C.; Liang J. Variations in oral microbiota associated with oral cancer. Sci Rep. 2017, 7, 1-10.
[25]- Perera, M.; Al-Hebshi, N.N.; Speicher, D.J.; Perera, I.; Johnson, N.W. Emerging role of bacteria in oral carcinogenesis: a review with special reference to perio-pathogenic bacteria. J Oral Microbiol. 2016, 26, 1-10.
8/57: check grammar
Response
We have made the correction see line 74.
9/58: which ones? give examples
Response
Please see lines 74 to 81.
10/59: which ones? give examples
Response
Please see lines 74 to 81.
11/60: which species cause which disease? be more specific
Response
We gave some examples line 83 to 86 and we have added 2 citations and listed them in the references section.
[27] Eberhard, J; Stumpp, N; Winkel, A; Schrimpf, C; Bisdas, T; Orzak, P; Teebken, O.E; Haverich, A; Stiesch, M. Streptococcus mitis and Gemella haemolysans were simultaneously found in atherosclerotic and oral plaques of elderly without periodontitis-a pilot study. Clin Oral Investig. 2017, 21, 447-452.
[28] Fan, X; Alekseyenko, A.V; Wu, J; Peters, B.A; Jacobs, E.J; Gapstur, S.M; Purdue, M.P; Abnet, C.C; Stolzenberg-Solomon, R; Miller, G; Ravel, J; Hayes, R.B; Ahn, J. Human oral microbiome and prospective risk for pancreatic cancer: a population-based nested case-control study. Gut. 2018, 67, 120-127.
12/62: which metabolites?
Response
We gave more precisions lines 90 to 96.
13/ 63: how?
Response
We explain more this point lines 90 to 96 and we have added 2 citations and listed them in the references section.
[30] Flemming, H.C.; Wingender, J.; Szewzyk, U.; Steinberg, P.; Rice, S.A.; Kjelleberg, S. Biofilms: an emergent form of bacterial life. Nat Rev Microbiol. 2016, 14, 563-575.
[31] Holt, S.C.; Ebersole, J.L. Porphyromonas gingivalis, Treponema denticola, and Tannerella forsythia: the "red complex", a prototype polybacterial pathogenic consortium in periodontitis. Periodontol 2000. 2005, 38, 72-122.
14/64-65: give examples, which degrees linked to which microbial composition
Response
We gave some examples of microbial composition (lines 93 to 101). As these bacteria are essentially related to the pathogenesis and evolution of periodontal diseases, we have corrected our sentence to clarify our thinking. (Line 97 to 99).
15/69: check grammar
Response
Thank you for pointing this error. We corrected it
16/69: contradictory to previous sentence
Response
The sentence has been reworded to correct its contractibility. The new sentence is:
“The oral microbiome, therefore, appears to be crucial for health as it can cause both oral and systemic diseases”. Please see line 107.
17/2.1: the search terms are very restrictive: "microbiota" and "microbiome" are missing, synonyms for mental disorders and specific diseases should be included.
Response
We agree with the reviewer, our search equation may seem restrictive at first glance. However, it has been determined following an analysis of several terms relevant to our review (Mouth, microbiota, dysbiosis, oral health, mental disorders, brain diseases, neurocognitive disorders, periodontitis...). The correct adaptation of these terms to our work was identified using the mesh database of MEDLINE (Pubmed). The terms retained in our equation were selected thanks to a preliminary analysis in order to limit redundancies between them and the article selection they induce.
Although the term microbiome has been considered, it is not directly referenced in the Mesh terms database, it is associated with the keyword "Microbiota". In the further construction of our equation, the term "Microbiota" appeared to be inappropriate because it is too restrictive. Thus, when it was combined with the other terms used to define the oral microbiome (Mouth, Mouth Diseases, Saliva) in mentally ill patients with only 20 articles were identified, 7 of which were relevant to the criteria of our work which was also selected by our final equation. The use of the subheading /Microbiology appeared less limiting in our equation and was therefore used.
Similarly, the term "mental disorders" has emerged as the most comprehensive term for our selection of articles. Its use is associated with many synonyms (psychiatric diseases; psychiatric illnesses; psychiatric disorders; psychiatric diagnosis; behavioural disorders; severe mental disorders). The mesh tree it induces allows the identification of "psychiatric illnesses or diseases manifested by failures in the adaptation process expressed mainly by abnormalities in thinking, feeling and behavior producing either distress or impaired function" as defined in PubMed ( Anxiety Disorders ; Bipolar and Related Disorders; Dissociative Disorders; Eating and Nutrition Disorders; Mood Disorders; Neurocognitive Disorders; Neurodevelopmental Disorders; Schizophrenia Spectrum and Other Psychotic Disorders; Sleep and Alertness Disorders. ..).
18/86: why restricted to 2000?
Response
Although the first work on the intestinal microbiota dates back to the beginning of the century, the development of high-throughput sequencing techniques for genetic material in the 2000s gave new impetus to this research. This is why we started our research in 2000 in order to focus on the most relevant work. As far as the study of the brain-gut relationship is concerned, the year 2004 is often cited as the starting point with reference to the work of Sudo et al. 2004.
We clarified this point in the « Search strategy » section (lines 125 to 127) to the manuscript and we added a new reference
[40] Sudo, N; Chida, Y; Aiba, Y; Sonoda, J; Oyama, N; Yu, XN; Kubo, C; Koga, Y. Postnatal microbial colonization programs the hypothalamic-pituitary-adrenal system for stress response in mice. J Physiol. 2004, 558, 263-275
19/2.2.1: give examples for dysbiosis
Response
We explained this point lines 141 to 145 and we have added 5 references which have been listed in the references section.
[41] Petersen, C.; Round, J.L. Defining dysbiosis and its influence on host immunity and disease. Cell Microbiol. 2014, 16, 1024–1033.
[42] Bäckhed, F.; Fraser, C.M.; Ringel, Y.; Sanders, M.E.; Sartor, R.B.; Sherman, P.M.; Versalovic, J.; Young, V.; Finlay, B.B. Defining a healthy human gut microbiome: current concepts, future directions, and clinical applications. Cell Host Microbe. 2012, 12, 611-622.
[43] Trompette, A.; Gollwitzer, E.S.; Yadava, K.; Sichelstiel, A.K.; Sprenger, N.; Ngom-Bru, C.; Blanchard, C.; Junt, T.; Nicod, L.P.; Harris, N.L.; Marsland, B.J. Gut microbiota metabolism of dietary fiber influences allergic airway disease and hematopoiesis. Nat Med. 2014, 20, 159-166
[44] Hsiao, E.Y.; McBride, S.W.; Hsien, S.; Sharon, G.; Hyde, E.R.; McCue, T.; Codelli, J.A.; Chow, J.; Reisman, S.E.; Petrosino, J.F.; Patterson, P.H.; Mazmanian, S.K. Microbiota modulate behavioral and physiological abnormalities associated with neurodevelopmental disorders. Cell. 2013, 155, 1451-1463.
[45] Garrett, W.S. Cancer and the microbiota. Science. 2015, 348, 80–86
20/Fig. 1: not clear why the search terms are so restrictive (see above)
Response
We explained this point in detail in our response to question 17 above.
21/Fig. 1: use English expressions for diseases
Response
Thank you for pointing this error. We corrected it
22/Fig. 1: what is meant by "out of equation"?
Response
This means that we have added 3 more items and they are outside of our search equation. We removed "out of equation" and added "Article found during the secondary search ". We clarified this point in the Data collection (lines 150 to 152) and Results of study selection sections, lines 158 to 160. We hope this will be clearer.
23/Fig. 1: AD is overrepresented (59.1%), number of publications for other diseases are quite low (1,3,5).
Response
We agree, we discussed this point in the discussion section
24/115-116: objectives, results, limitations are missing in Table 1.
Response
As requested, we have introduced in Table 1 a column specifying the objectives of the studies. We have also modified the summary column to present the results and limitations of the studies. We thank the reviewer for this suggestion.
25/3.5. This section is based on only 1 article
Response
We agree, we discussed this point in the discussion section.
26/Table 1: some interesting information of the references is missing (see comment 115-116), furthermore the involved species in an extra-column and the suggested route of action would add significant value to the overview
Response
As you suggested, we have identified the bacterial species in an additional column inserted in Table 1.
Mechanisms of action that may introduce the notion of a relationship between the oral microbiome and the brain are only rarely suggested in studies. To underline this point, we have modified the discussion section lines 370 to 379.
We have added 1 citation and listed them in the references section.
[91] Carabotti, M; Scirocco, A; Maselli, M.A, Severi, C. The gut-brain axis: interactions between enteric microbiota, central and enteric nervous systems. Ann Gastroenterol. 2015 ,28, 203-209.
27/Table 1: Sun 2020: what is meant by "not provide convincing evidence" - be more precise. what is claimed in the paper?
Response
Thank you for highlighting this unclear element. We have modified Table 1 to make it clear that this study cannot conclude that there is a causal link between Alzheimer's disease and periodontal disease based on the analysis of data from the Genome-Wide Association Studies on periodontitis and AD.
28/Table 1: Poole 2013: Lipopolysaccharide
Response
To be more precise, we remove the lipopolysaccharide of the periodontal bacteria and replace it by Porphyromonas gingivalis (Pg). We hope that this will be clearer.
29/3.2.1 and 3.2.2 are confusing. Why are they separated? S. mutans is a caries associated species - why is it listed in the perio section?
Response
We agree. Titles 3.2.1 and 3.2.2 induce confusion by suggesting that periodontal disease is only related to Alzheimer's disease, dementia and cognitive disorders. We have deleted these two titles and reordered this chapter.
30/Discussion should dig deeper into the findings and interconnect them.
Response
As requested, we have made major changes to the discussion section.
31/Conclusion should provide more details and the potential of the topic.
Response
We have revised the conclusion section taking into account your suggestions.

Reviewer 2 Report
The systematic review submitted by Maitre et al aimed to investigate the role of the oral microbiome in the pathophysiology of mental health disorders and to introduce the oral-brain axis as a new research perspective. Indeed this is a research area with growing interest and deserves review. The primary critiques:
- The oral-brain axis is not a new research perspective, but instead an area with growing interest.
- Key references are missing, likely due to the search terms which did not include the key word “oral”. Unclear why “mouth” and not “oral” was used a primary search term. This is a major oversight and flaw in the search criteria given that the aim of this study was to “investigate the role of the oral microbiome in the pathophysiology of mental health disorders”.
- Inappropriate grouping of brain disorders.
- Alzheimer’s disease (AD) is not a mental disorder, it is a disease defined by the neurobiology which include amyloid beta plaques and tau-containing neurofibrillary tangles.
- Periodontal disease and the oral microbiome in individuals with Down syndrome (DS) could be linked to trisomy 21, therefore it is inappropriate to apply generalizations from individuals with DS to mental retardation in general, e., DS has a clear genetic cause which could influence the oral microbiome distinct from non-DS associated mental retardation.
- Hepatic encephalopathy (line 152) should not be grouped with Alzheimer’s disease and cognitive dysfunction.
Minor comments:
- Line 69, delete “is”, sentence should read, “The oral microbiome, therefore, appears…”
- The paragraph starting on Line 76 should be combined with the prior paragraph.
- Figure 1 is not in English
Author Response
Response to reviewers 2
We thank the Reviewer for giving us an opportunity to substantially improve the content and the presentation of our manuscript. We have modified the article in accordance with your requests. You will find every modification in the text using track changes, and the pages are noted in the answer for every point below.
Reviewer 2
The systematic review submitted by Maitre et al aimed to investigate the role of the oral microbiome in the pathophysiology of mental health disorders and to introduce the oral-brain axis as a new research perspective. Indeed, this is a research area with growing interest and deserves review. The primary critiques:
1/The oral-brain axis is not a new research perspective, but instead an area with growing interest.
Response
We have changed the sentence “new research perspective” to “growing interest area” in the abstract, line 17 and at the end of the introduction, line 119 and in the conclusion section, line 422.
2/Key references are missing, likely due to the search terms which did not include the key word “oral”. Unclear why “mouth” and not “oral” was used a primary search term. This is a major oversight and flaw in the search criteria given that the aim of this study was to “investigate the role of the oral microbiome in the pathophysiology of mental health disorders”.
Response
While the term "oral" may appear obvious as the main search term, it is not referenced in the Mesh directory of the Medline database (Pubmed).
The use of the search term "Mouth" including many synonyms (Oral Cavity; Cavitas Oris; Vestibule of the Mouth; Vestibule Oris; Oral Cavity Proper; Mouth Cavity Proper; cavitas oris propria) as well as a tree structure defining all the microbial compartments of the oral cavity (dentition, oral mucosa, palate, tongue...), it appears adapted to our work. The use of this single term can be restrictive, so it has been associated with the terms "Mouth diseases" and "Saliva" to broaden our search for relevant references to specific microbiomes associated with secretions and disorders in the oral cavity.
3/Inappropriate grouping of brain disorders.
Alzheimer’s disease (AD) is not a mental disorder, it is a disease defined by the neurobiology which include amyloid beta plaques and tau-containing neurofibrillary tangles.
Response
Alzheimer’s disease (AD) and its symptoms are outlined in the Diagnostic and Statistical Manual of Mental Disorders fifth edition) (DMS V), which is the main tool used to diagnose mental illnesses such as Schizophrenia and Borderline Personality Disorder. In this context, AD is formally recognized as a mental illness.
ICD 10 also identifies Alzheimer's disease as a mental disorder since it appears in Chapter V corresponding to “Mental and behavioural disorders” in which it is included in the group of organic mental disorders (F00).
As you rightly pointed out, these classifications clearly incorporate the notion of the neurodegenerative origin of Alzheimer's disease, which is still identified to this day as a mental disorder because of its multiple cognitive consequences.
To clarify this point, we introduced this sentence line 177 to 180.
“AD and its symptoms although origin is neurodegenerative are outlined in the Diagnostic and Statistical Manual of Mental Disorders fifth edition (DMS V) and In the International Classifications of Diseases, 10th Revision (ICD-10) and is recognized as a mental illness with a neurological origin.
4/Periodontal disease and the oral microbiome in individuals with Down syndrome (DS) could be linked to trisomy 21, therefore it is inappropriate to apply generalizations from individuals with DS to mental retardation in general, e., DS has a clear genetic cause which could influence the oral microbiome distinct from non-DS associated mental retardation.
Response
Just like you, we are well aware of the specific immune character attached to trisomy 21. Unfortunately, studies of oral microbiome analysis in DS patients generally use matching with non-DS patients with mental retardation and/or healthy patients. This has forced us to associate the results of DS and non-DS patients with mental retardation.
In order to highlight this important point, we have clarified this confounding factor in the “discussion” section, line 350 to 357 and “limitations” chapter line 398 to 399.
5/Hepatic encephalopathy (line 152) should not be grouped with Alzheimer’s disease and cognitive dysfunction.
Response
Residual cognitive impairment associated with hepatic encephalopathy having, like Alzheimer's disease, an organic origin, we have decided to associate it in the chapter dealing with Alzheimer's disease, dementia and cognitive disorders. In order to reduce the association between hepatic encephalopathy and Alzheimer's disease, we have modified the presentation of this study. Please see the change, line 220 to 221 in the “results” section.
Minor comments:
6/Line 69, delete “is”, sentence should read, “The oral microbiome, therefore, appears…”
Response
Thank you for pointing this error. We deleted ‘’is”.
7/The paragraph starting on Line 76 should be combined with the prior paragraph.
Response
We have made the requested change.
8/Figure 1 is not in English
Response
Thank you for pointing this error. We have corrected.

Round 2
Reviewer 1 Report
With the current adaptations the manuscript was significantly improved. The additional informations in Table 1 contributes essentially to the understanding of the review and the added references makes the introduction and discussion stronger.
Considering the following minor changes.
17: a growing interest area
29/30: rephrase, not clear.
81, 83, : remove points in species names
89: hunter-gatherers vs. Westerners: be more precise which populations were compared?
89/90: do not remove "can" and "the mouth"
97: "closely related" is too weak - i suggest a more causative expression. the microbiota directly influences those diseases.
102: connection to systemic diseases not clear, last sentence was about perio. make new paragraph when writing about other diseases.
105: the oral microbiome
107/108: cause systemic disease: this is not described before - a better description would be:
- causative for oral diseases (examples given before)
- change in oral microbiota observed in systemic diseases
119: delete
150: inserted sentence not clear
Figure 1: be more precise in the figure legend. flow chart of what?
Table 1: Insertion of additional columns significantly improved this summary table
column F: Main results
D2: Investigation of the subgingival (2x)
D3: rephrase "determine whether"
F3: was Treponema measured in brain tissue? if yes, rephrase: e.g. Detection of oral Terponema
ganglia of the brainstem
Post mortem contaminations cannot be excluded
D4: Comparison
F4: controls
D5: Testing of the
antibodies against periodontal
what do you mean by "more important"? higher concentration?
rephrase last sentence: e.g. investigation of TNF-alpha and antibodies as diagnostic biomarkers for AD
F5: "associate with" is not clear - rephrase
Small sample size
D6: Assessment of the relationship between
F6: "common periodontitis pathogen" - specify: which biomarker was measured in blood?
D7: Investigation of
F7: Down syndrome
D8: rephrase "establish"
F8: add a limitation
D9: Shorten: first half-sentence is duplicate of next one. if sub-objectives use one line for each
F9: immune status
D10: Investigation of
D11: Characterization of
E11: taxa
F11: induced
D12: Examination of
270: delete "a mere"
272: depside?
279: English
285: oral
290: bacteria
292: "predominant" is misleading, describe more specific (e.g. increase of bacterial load, elevated inflammatory markers).
319: rephrase "the vary, replicated"
322: remove "to"
369: allow a discrimniation between
373: modulate the oral
419: what is meant by "sites"?
424: research
427: literature represents a basis
Author Response
Response to reviewer 1
We thank the Reviewer for giving us an opportunity to substantially improve the content and the presentation of our manuscript. We have modified the article in accordance with your requests. You will find every modification in the text using track changes (in green color), and the pages are noted in the answer for every point below.
Reviewers 1
With the current adaptations the manuscript was significantly improved. The additional informations in Table 1 contributes essentially to the understanding of the review and the added references makes the introduction and discussion stronger.
Response
Tank you
Considering the following minor changes.
17: a growing interest area
Response
We have made the correction.
29/30: rephrase, not clear.
Response
We rephrase this sentence to “Researchers should focus their efforts to develop research on the oral-brain axis in the future.” (Line 30 to 31).
81, 83 : remove points in species names
Response
We have made the correction.
89: hunter-gatherers vs. Westerners: be more precise which populations were compared?
Response
Please see line 90 to 91.
89/90: do not remove "can" and "the mouth"
Response
We have made this correction line 92.
97: "closely related" is too weak - i suggest a more causative expression. the microbiota directly influences those diseases.
Response
Thank you for this suggestion. We have introduced this sentence: The oral microbiota directly influences dental caries and periodontal diseases (line 100).
102: connection to systemic diseases not clear, last sentence was about perio. make new paragraph when writing about other diseases.
Response
We agree. We revised the paragraph line 105 to 112 to make it clearer.
105: the oral microbiome
Response
We made the correction (Line 113 in the manuscript).
107/108: cause systemic disease: this is not described before - a better description would be:
- causative for oral diseases (examples given before)
- change in oral microbiota observed in systemic diseases
Response
We thank the reviewer for this suggestion. We have made change as requested lines 107 to 108.
119: delete
Response
We deleted “which is a growing interest area” as requested (Line 126).
150: inserted sentence not clear
Response
We removed “A complementary analysis of the titles and abstracts of similar articles associated in the PubMed database with the articles thus selected to completed the selection. “and replaced to “A scan of the references of the previously selected articles completed the selection” line 159.
Figure 1: be more precise in the figure legend. flow chart of what?
Response
We introduced “of included studies “to be more precise.
Table 1: Insertion of additional columns significantly improved this summary table
Response
Thank you
column F: Main results
Response
We apologize, but during the submission process (R1), part of Table 1 was not registered. On the other hand, the pdf version associated with the submitted version was complete? We introduced the missing parts of the table and took into account all your remarks.
D2: Investigation of the subgingival (2x)
Response
We have made the suggested correction.
D3: rephrase "determine whether"
Response
To be clearer we reformulated the sentence in D3 section: “Assessment of the oral treponema presence in the human brain, the hippocampus and the trigeminal ganglia in AD patients and controls.”
F3: was Treponema measured in brain tissue? if yes, rephrase: e.g. Detection of oral Terponema
ganglia of the brainstem Post mortem contaminations cannot be excluded
Response
The aim of this study was to determinate and characterize the presence of oral treponema in frontal lobe cortex, hippocampus and trigeminal ganglia from AD patients and controls. As you suggested, we reformulated the mains results and the limitation to be clearer in the F3 section:
“Presence of oral Treponema in the trigeminal ganglia, the brainstem, and the cortex of human subjects with AD.”
“AD patients were more likely to have Treponema than controls, and they had more different Treponema species in brain than controls.”
“*Postmortem contamination of tissues or assays cannot be excluded”
D4: Comparison
Response
We have made the suggested correction.
F4: controls
Response
Thank you for pointing this error. We corrected it.
D5: Testing of the antibodies against periodontal what do you mean by "more important"? higher concentration?
Response
The authors test the hypothesis that AD patients have a higher concentration of cytokines and antibodies against P. gingivalis, T. forsythia and A. actinomycetemcomitans than non-AD patients. So “more important” means more elevated concentration. To clarify this point, we modified the sentence in the D5 section: “Testing the hypothesis that concentration of TNF α and antibodies raised to periodontal bacteria would be more elevated in AD compared to normal controls.”
Rephrase last sentence: e.g. investigation of TNF-alpha and antibodies as diagnostic biomarkers for AD
Response
We have made the suggested correction.
F5: “associate with” is not clear – rephrase Small sample size
Response
We reformulated the sentence to be clearer and made the modification suggested for the limitation.
“AD patients have both elevated cytokine expression and antibodies to periodontal bacteria.
These measures could contribute to the diagnosis of AD.”
D6: Assessment of the relationship between
Response
We have made the suggested correction.
F6: "common periodontitis pathogen" - specify: which biomarker was measured in blood?
Response
As requested, we specified the biomarker measured: Pg Immunoglobulin G
D7: Investigation of
Response
We have made the suggested correction.
F7: Down syndrome
Response
We have made the suggested correction.
D8: rephrase "establish"
Response
We removed "establish a relation" and replace it by “Evaluation of the role”.
F8: add a limitation
Response
Methodological limitations were added (Hospital patient selection, study design, small sample size).
D9: Shorten: first half-sentence is duplicate of next one. if sub-objectives use one line for each
Response
We have modified as suggested.
F9: immune status
Response
We have made the suggested correction.
D10: Investigation of
Response
We have made the suggested correction in the D20 section of the complete table.
D11: Characterization of
Response
We have made the suggested correction in the D21 section of the complete table.
E11: taxa
Response
We have made the suggested correction in the E21 section of the complete table.
F11: induced
Response
Thank you for pointing this error. We corrected it in the F21 section of the complete table.
D12: Examination of
Response
We have made the suggested correction in the D22 section of the complete table.
270: delete "a mere"
Response
We deleted “a mere” as request (line 284).
272: depside?
Response
We replaced by “despite” (line 286).
279: English
Response
We reformulated the sentence to make it clearer: “The identification of a specific microflora in patients with mental disorders suggests that these disorders could influence the oral microbiome.” (Line 294).
285: oral
Response
We have made the correction (line 300).
290: bacteria
Response
We have made the correction (line 304).
292: "predominant" is misleading, describe more specific (e.g. increase of bacterial load, elevated inflammatory markers).
Response
We used your suggested correction to clarify our sentence Lines 305 to 307.
319: rephrase "the vary, replicated"
Response
As request we rephrase the sentence line 319. Please see lines 335 to 336 of the revised version.
322: remove "to"
Response
The correction was made, line 338.
369: allow a discrimination between
Response
The correction was made, line 385.
373: modulate the oral
Response
We deleted “of” please see line 390.
419: what is meant by "sites"?
Response
Thank you for pointing out this unclear part of the sentence, we have reworded it to make it clearer (Lines 436 to 437 and 439).
424: research
Response
The correction was done line 442.
427: literature represents a basis
Response
Thank you for this suggestion. We made the change line 445.
